# SARS-CoV-2 Antibody Response and Sustainability after a Third Dose of BNT162b2 in Healthcare Workers at Health Promotion Centers

**DOI:** 10.3390/v15030751

**Published:** 2023-03-14

**Authors:** Eun-Hee Nah, Seon Cho, Hyeran Park, Suyoung Kim, Dongwon Noh, Eunjoo Kwon, Han-Ik Cho

**Affiliations:** 1Department of Laboratory Medicine and Health Promotion Research Institute, Korea Association of Health Promotion, Seoul 07572, Republic of Korea; 2MEDIcheck LAB, Korea Association of Health Promotion, Seoul 07572, Republic of Korea

**Keywords:** third dose, vaccination, SARS-CoV-2, anti-spike-protein receptor-binding domain (S-RBD) antibodies, mRNA vaccines

## Abstract

The aim of this study was to determine the antibody response and the sustainability of immunogenicity after a third dose of BNT162b2 (BNT) in homologous [ChAdOx1 (ChAd)/ChAd, BNT/BNT, and mRNA-1273 (Moderna)/Moderna] and heterologous (ChAd/BNT) vaccinations of two primary doses with different schemes. This prospective observational study recruited consenting healthcare workers from 16 health checkup centers in 13 Korean cities. Three-point blood tests were analyzed as the antibody response after the third vaccination: T3-1 (1 month after the third dose), T3-3 (3 months after the third dose), and T3-4–10 (4–10 months after the third dose). SARS-CoV-2 antibodies were measured using a chemiluminescence microparticle immunoassay with SARS-CoV-2 IgG II Quant in the ARCHITECT system (Abbott Diagnostics). The antibody levels were significantly higher in the Moderna /Moderna and BNT/BNT groups than in the ChAd/ ChAd and ChAd/BNT groups (*p* < 0.05) at T3-1. At T3-3, antibody levels had decreased by 29.1% in the BNT/BNT group and by 45.3% in the ChAd/ChAd group compared with the antibody levels at T3-1. The anti-SARS-CoV-2 S-RBD IgG levels at T3-1 were significantly associated with having received mRNA vaccines as the two primary doses (*p* < 0.001). The third dose of BNT induced an increased humoral immune response in various vaccination schemes, which was more prominent for the two primary doses of homologous mRNA vaccines. However, this immunogenicity decreased within 3–10 months after the third dose. These results suggest that another booster dose (a fourth dose), which would be able to counteract SARS-CoV-2 variants, is needed.

## 1. Introductions

There is strong evidence that the two primary doses of vaccines offer substantial protection against coronavirus disease 2019 (COVID-19), but antibody titers decline relatively rapidly after receiving the second dose [1,2,3]. Furthermore, the enhanced transmission of the delta (B.1.617.2) variant that began in the spring of 2021 resulted in increased numbers of breakthrough infections in fully vaccinated people [4]. The omicron (B.1.1.529) wave quickly followed in November 2021 [5]. A third vaccine dose has therefore been recommended for not only immunocompromised patients but also eventually the entire population older than 12 years.

Booster (third) doses enhance waning immunity and expand the breadth of immunity against severe acute respiratory syndrome coronavirus 2 (SARS-CoV-2) variants of concern [6,7]. Third doses provide rapid and substantial increases in protection against both mild and severe diseases [8,9]. Studies have found that booster doses of messenger RNA (mRNA)-based vaccines decreased the occurrence of documented infections during periods with delta-variant predominance [10,11]. 

The measurement of binding and neutralizing antibody levels can be useful in predicting efficacy after boosting since they are correlated with the efficacy of both mRNA and adenovirus-vectored vaccines [12,13]. Some studies have demonstrated immunologic advantages for homologous and heterologous prime-boost strategies [14,15]. Homologous booster vaccines of BNT162b2 (BNT) and mRNA-1273 (Moderna) were found to increase neutralizing antibody titers against the wild-type SARS-CoV-2 virus, and against the delta variant [16,17]. However, there have been few reports of the sustainability of the immunogenicity among various primary vaccination schemes. The aims of this study were to determine the antibody response and measure the sustainability of immunogenicity after a third BNT dose in homologous [ChAdOx1 (ChAd)/ChAd, BNT/BNT, and Moderna/Moderna] and heterologous (ChAd /BNT) vaccinations of two primary doses in different vaccination schemes among healthcare workers (HCWs) at health promotion centers in South Korea.

## 2. Materials and Methods

### 2.1. Study Design and Participants

The present study involved a cohort constructed in our previously published study [18]. The inclusion criteria were as follows: HCW with a vaccination plan and consented HCW. Subjects who had COVID-19-related symptoms at the time of the study or pregnancy were excluded. HCWs in 16 health promotion centers received a third dose between November 2021 and February 2022, a period of delta- and omicron-variant predominance. This prospective observational study recruited 869 consenting HCWs who had already received two vaccine doses at least 4 months before their third BNT dose. On the previous two vaccine doses, HCWs had been tested for anti-S-RBD IgG antibodies at T0 (day of first dose), T1-1 (1 month after first dose), T2-0 (day of second dose), T2-1 (1 month after second dose), and T2-3 (3 months after second dose). The findings of blood tests performed at three time points were analyzed as the antibody response after the third vaccination: T3-1 (1 month after the third dose), T3-3 (3 months after the third dose), and T3-4–10 (4–10 months after the third dose).

This study was approved by the Institutional Review Board of the Korea Association of Health Promotion (approval no. 130750-202104-BR-001). Each participant signed a written informed consent form.

### 2.2. Receptor-Binding Domain (RBD) IgG Measurement for Anti-SARS-CoV-2 Spike Protein

The SARS-CoV-2 IgG II Quant assay (Abbott Diagnostics, Chicago, IL, USA) is a chemiluminescence microparticle immunoassay used for the qualitative and quantitative determination of IgG SARS-CoV-2 antibodies in human serum on the ARCHITECT i System (Abbott, Lake Forest, IL, USA) [19]. This assay was designed to detect SARS-CoV-2 IgG RBD and neutralizing antibodies in serum. The ancestral RBD was utilized for the assay. Plaque reduction neutralization (PRNT) is used to quantify the titer of neutralizing antibodies for a virus. A positive percent agreement study was performed with the SARS-CoV-2 IgG II Quant assay that was demonstrated to be positive (≥1:20) using a PRNT by the Broad Institute. It utilizes a Y-weighted four parameter logistic curve data-fit reduction method to generate calibrations and results. The cutoff value for a positive result was defined as ≥50 AU/mL (values < 50 AU/mL were considered negative) [20]. The lower limit of quantification was 21.0 AU/mL, according to information from the manufacturer. The measurement range was 21.0–40,000 AU/mL, and values above this range were recorded as 40,000 AU/mL [21].

### 2.3. Statistical Analyses

Statistical analyses were performed using SAS version 9.4 (SAS Institute, Cary, NC, USA). Demographic characteristics were presented as number (percentage) values. Data are presented as mean ± standard deviation, and frequency (percentage) values. Differences in characteristics between the four primary vaccine groups were analyzed using one-way ANOVAs with post-hoc (Scheffe’s method) and chi-square tests. The differences in antibody levels before and after the third vaccine dose were analyzed using a paired *t*-test. The differences in antibody levels between T3-1 and T3-3 were analyzed using paired *t*-tests. The differences in the antibody levels among the four primary vaccine groups at each blood test time point were analyzed using one-way ANOVAs with a post-hoc test (Scheffe’s method). The log-transformed values were used in regression models for anti-SARS-CoV-2 S-RBD IgG levels with a skewed distribution. Univariable and multivariable liner regression analyses were performed to verify the associated factors for anti-SARS-CoV-2 S-RBD IgG levels 1 month after the third dose. *p* values of <0.05 were considered significant.

## 3. Results

### 3.1. Characteristics of the Study Participants

Overall, 869 participants who received two primary vaccine doses participated in this study: ChAd/ChAd, ChAd/BNT, BNT/BNT, and Moderna/Moderna vaccines were administered to 25, 581, 206, and 57 participants, respectively. The mean age was lowest in the BNT/BNT group (33.9 years), and the intervals between the second and third vaccines were the shortest in the Moderna/Moderna group (*p* < 0.001) (Table 1).

### 3.2. Antibody Levels and Their Sustainability after the Third Dose

Anti-SARS-CoV-2 spike (S)-RBD IgG levels had increased at 1 month after the third dose in all four groups (*p* < 0.001) (Figure 1). Anti-SARS-CoV-2 S-RBD IgG levels were significantly higher in the BNT/BNT and Moderna/Moderna groups than in the ChAd/ChAd and ChAd/BNT groups at T3-1 (*p* < 0.001) (Figure 2). At T3-3, the percent reduction of anti-SARS-CoV-2 S-RBD IgG levels was analyzed in each group compared to T3-1. At T3-3, antibody levels had decreased by 29.09% in the BNT/BNT group compared with 45.29% in the ChAd/ChAd group (Figure 3). At the maximum follow-up period of 10 months, a 97.0% reduction in anti-SARS-CoV-2 S-RBD IgG levels occurred in all groups (data not shown). The mean antibody levels before and after all three vaccine doses were significantly higher in the Moderna/Moderna and BNT/BNT groups than in the ChAd/ChAd group (*p* < 0.001) (Figure 4). 

### 3.3. Factors Associated with Anti-SARS-CoV-2 S-RBD IgG Levels at T3-1

Female sex was associated with anti-SARS-CoV-2 S-RBD IgG levels in the univariable analyses (*p =* 0.016), but this association disappeared in the multivariable linear regression analysis (*p =* 0.098). Multivariable linear regression analysis indicated that anti-SARS-CoV-2 S-RBD IgG levels at T3-1 were significantly associated with having received mRNA vaccines as the two primary doses (*p* < 0.001) (Table 2).

## 4. Discussion

This prospective observational study estimated the antibody response and persistence of immunogenicity after a third dose of BNT in homologous (ChAd/ChAd, BNT/BNT, and Moderna/Moderna) and heterologous (ChAd/BNT) vaccinations of two primary doses in different vaccination schemes. The third BNT vaccine dose was administered on average 4 months after the second dose and induced increased humoral responses in the various vaccination schemes, which were more prominent when the primary two doses were homologous mRNA vaccines. However, this immunogenicity decreased 3–10 months after the third vaccination.

The third vaccine dose was immunogenic in all four primary vaccine schemes in the present study, which was consistent with previous findings [22,23]. The antibody titer after homologous Moderna vaccine boosting was substantially higher in those who received the two primary Moderna vaccines than in those who only received two primary Moderna doses with heterologous booting [24]. This finding indicated a robust anamnestic response after booster vaccination. Furthermore, some studies have demonstrated that mRNA booster vaccines increase antibodies and substantially improve protection against COVID-19 [22,25]. However, these increased antibody levels decreased as time went on. Despite a progressive decline over time, booster mRNA vaccines may lead to high titers of neutralizing antibodies, which may provide greater protection against symptomatic infection with variants [12]. In the present study, a reduction in the increased antibody levels was also observed 3 months after the third vaccine dose in the present study. At the maximum follow-up period of 10 months, although there was no seronegative conversion, participants in all groups presented an approximate 97.0% reduction in anti-SARS-CoV-2 S-RBD IgG levels.

The participants in the present study received different vaccination schemes in the primary course. ChAd was the most common vaccine used early in the course for HCWs as the clinical risk group. The mRNA vaccines were the most common vaccines used in persons younger than 40 years after an association between ChAd and vaccine-induced thrombotic thrombocytopenia was reported [26]. Several studies compared the immunogenicity of heterologous and homologous prime-boost vaccinations [14,15,23]. Regarding immunogenicity, prebooster antibody levels were similar or higher after heterologous boosting (i.e., different from the primary vaccine) than after homologous boosting [23]. The heterologous boost immunization strategy provides an immune response that may be beneficial for the durable prevention and control of COVID-19 [27,28]. On the other hand, a study found that a homologous Moderna/Moderna group had higher anti-RBD levels from 2 weeks after the first vaccine and 14–20 days after the third vaccine relative to the heterologous ChAd/Moderna group; however, these differences became less prominent after the third dose [29]. Our finding that anti-SARS-CoV-2 S-RBD IgG levels were significantly higher in the BNT/BNT and Moderna/Moderna groups than in the ChAd/ChAd and ChAd/BNT groups at T3-1 was consistent with the findings of that previous study. We previously found that anti-S-RBD levels were the highest in participants who received homologous mRNA vaccines as the two primary doses [18]. The antibody response after the third vaccine dose was consistent with that of the two primary doses of the vaccine program in the present study. The multivariable linear regression analysis indicated that anti-SARS-CoV-2 S-RBD IgG levels at T3-1 were significantly associated with having received two primary doses of mRNA vaccines after adjusting for age, sex, and the interval between the second and third doses.

This study has some limitations. First, the immunogenicity data were limited to antibody responses, and anti-SARS-CoV-2 S-RBD IgG antibodies were used as neutralizing antibody surrogates in this study. Although vaccine-induced spike-specific CD4+ and CD8+ T-cell responses may contribute to the durability of the antibody response and the prevention of severe disease [27,28], we only evaluated the anti-SARS-CoV-2 S-RBD IgG levels after the third vaccination dose, which would compromise the study conclusions. Second, a small number of participants remained after T3-3, especially at 9 months after the third dose, which restricted the ability to verify antibody durability after the third vaccination dose. Third, the study participants were HCWs who only included a small proportion of subjects older than 60 years, which might not represent the general population. Furthermore, there is a large difference in the number of participants in each group, and the groups are also not homogeneous in terms of gender and age. This study was not a randomized control study but a prospective observational study in a consented HCW cohort. In Korea, there were vaccine supply shortages and interruptions at the start of vaccination. The Korean government recommended various vaccine types and cross-platform mixed-dosing strategies according to vaccine supply. Koreans could not choose vaccine types. Nevertheless, authors tried to adjust for age, sex, and vaccine types when analyzing data to overcome this limitation.

In conclusion, this study has characterized the antibody response and persistence of the immunogenicity after a third BNT dose and how long the increased antibody response from the third dose is maintained in various vaccination schemes. These results suggest that another improved booster dose (a fourth dose), which would be able to counteract SARS-CoV-2 variants, is needed.

## Figures and Tables

**Figure 1 viruses-15-00751-f001:**
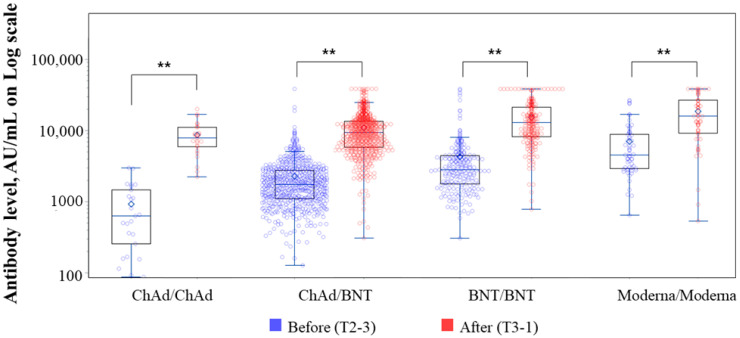
Anti-SARS-CoV-2 S-RBD IgG levels before (T2-3) and after (T3-1) the third vaccine in each group. Abbreviations: (T2-3), 3 months after the second dose; (T3-1), 1 month after the third dose. ** *p*-values (*p* < 0.001) were calculated using the paired *t*-test.

**Figure 2 viruses-15-00751-f002:**
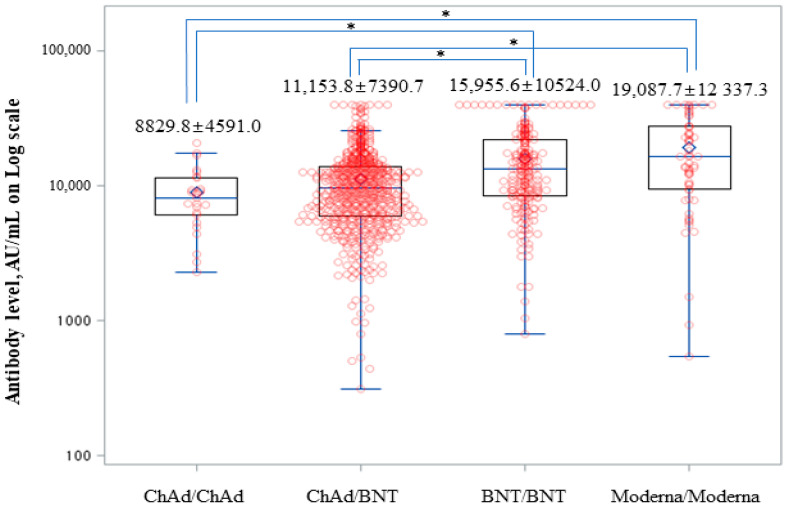
Anti-SARS-CoV-2 S-RBD IgG levels at T3-1 by vaccination type (*p* < 0.001). * *p*-values (*p* < 0.01) were calculated by a post-hoc analysis using Scheffe’s test: ChAd/ChAd, ChAd/BNT < BNT/BNT, and Moderna/Moderna.

**Figure 3 viruses-15-00751-f003:**
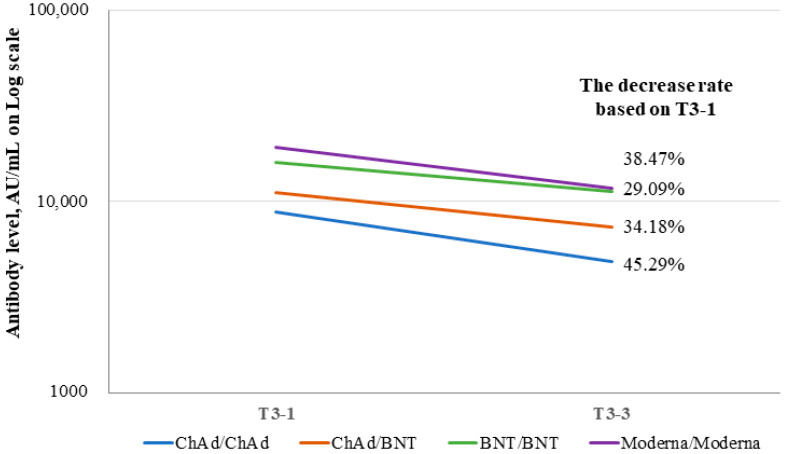
Change in anti-SARS-CoV-2 S-RBD IgG levels between 1 (T3-1) and 3 (T3-3) months after the third vaccine in each group. Abbreviations: T3-1 (1 month after the third dose); T3-3 (3 months after the third dose). *P*-values (*p* < 0.001) were obtained using paired *t*-tests in each group.

**Figure 4 viruses-15-00751-f004:**
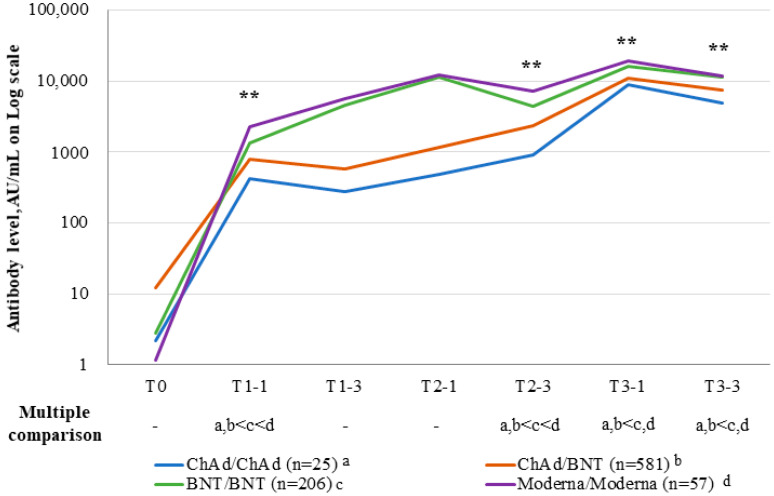
Mean anti-SARS-CoV-2 S-RBD IgG antibody levels for the four vaccination schemes before and after the vaccinations. T0 (day of first dose); T1-1 (1 month after the first dose); T1-3 (3 months after first dose); T2-1 (1 month after the second dose); T2-3 (3 months after the second dose); T3-1 (1 month after the third dose); T3-3 (3 months after the third dose). The mean difference among four vaccination schemes was tested using ANOVA at T1-1, T2-3, T3-1, and T3-3, respectively. ** *p*-values (*p* < 0.001) were calculated using an ANOVA test.

**Table 1 viruses-15-00751-t001:** Characteristics of the study subjects.

		Primary Vaccines								
		ChAd/ChAd		ChAd/BNT		BNT/BNT		Moderna/Moderna		*p*
		*N*	(%)	*N*	(%)	*N*	(%)	*N*	(%)	
**Total (*N* = 869)**		25		581		206		57		
**Sex**										
	Male	13	(52)	209	(36)	48	(23.3)	16	(28.1)	0.001
	Female	12	(48)	372	(64)	158	(76.7)	41	(71.9)	
**Age, years**										
	≤29	0	-	2	(0.3)	89	(43.2)	7	(12.3)	<0.001
	30–39	4	(16)	199	(34.3)	69	(33.5)	23	(40.4)	
	40–49	2	(8)	230	(39.6)	35	(17)	15	(26.3)	
	50–59	18	(72)	128	(22)	11	(5.3)	12	(21.1)	
	60–69	1	(4)	18	(3.1)	0	-	0	-	
	≥70	0	-	4	(0.7)	2	(1)	0	-	
	Mean ± SD	50.9	±9.2 ^a^	43.9	±8.6 ^b^	33.9	±9.1 ^c^	40.2	±9.4 ^d^	<0.001 ^†^
**Interval between second and third doses, months (mean ± SD)**		4.4	±0.7 ^a^	5.0	±0.8 ^b^	4.5	±1.1 ^c^	4	±0.9 ^d^	<0.001 ^‡^
**Working in patient-facing healthcare**										
	No	8	(32)	83	(14.3)	46	(22.3)	7	(12.3)	0.007
	Yes	17	(68)	498	(85.7)	160	(77.7)	50	(87.7)	

Abbreviation: SD, standard deviation. ^†^ Multiple comparisons among primary vaccines according to age: a > b > d > c. ^‡^ Multiple comparisons among primary vaccines according to the interval between the second and third vaccine doses: a < b, b > c > d.

**Table 2 viruses-15-00751-t002:** Factors associated with anti-SARS-CoV-2 S-RBD IgG levels 1 month after the third dose.

	1 Month after the Third Vaccination							
	Unadjusted				Adjusted			
	Coef.	(SE)	Exp (Coef.)	*p*-Value	Coef.	(SE)	Exp (Coef.)	*p*-Value
**Sex, ref: female**	−0.129	(0.053)	0.88	0.016	−0.087	(0.052)	0.92	0.098
Age	−0.004	(0.003)	-	0.100	0.004	(0.003)	-	0.187
Vaccine type, ref: ChAd/ChAd								
ChAd/BNT	0.159	(0.147)	1.17	0.279	0.16	(0.149)	1.17	0.283
BNT/BNT	0.497	(0.152)	1.64	0.001	0.536	(0.16)	1.71	0.001
Moderna/Moderna	0.609	(0.172)	1.84	<0.001	0.645	(0.176)	1.91	<0.001
Working in patient facing healthcare role, ref. no	−0.046	(0.067)	0.96	0.496	−0.019	(0.068)	0.98	0.783
Interval between the second and third doses, (months)	−0.031	(0.027)	-	0.250	0.027	(0.028)	-	0.342

Abbreviations: Coef., coefficient; SE, standard error; exp, exponential. The log-transformed values were used in regression models for anti-SARS-CoV-2 S-RBD IgG levels with a skewed distribution.

## Data Availability

The datasets generated during analyses and/or the current study are available from the corresponding author on reasonable request.

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
