# Peer review of "SARS-CoV-2 Antibody Response and Sustainability after a Third Dose of BNT162b2 in Healthcare Workers at Health Promotion Centers"

_viruses, 2023, doi:10.3390/v15030751_

Round 1

Reviewer 1 Report

The data presented follows up on previous work by this group characterizing antibody levels following various Covid vaccine strategies in a large Korean Healtcare worker cohort.  Specifically, this paper describes the finding of boosted antibody responses after a booster, or third vaccine dose.  

The paper is written well and is easy to follow, and the data presented logically. The overall manuscript could be improved by resolving a few issues:

1.  With regards to the study participants, there are no inclusion or exclusion criteria included in the materials. Perhaps they are listed in previous publications, but at the minimum, it needs to be clear if the study excluded participants with immune suppressing conditions (HIV, Diabetes, kidney disease, pregnancy) or medications. 

2. In section 2.2, it states that the study was designed to detect RBD and neutralizing antibodies in serum, yet the study did not in fact did not assess any functional capacity of the antibodies, such as neutralization. This needs to be clarified.  Also, in this section, it should be made clear that ancestral RBD is utilized for the assay.

3. In section 2.3, there is no description of the regression models used in the study, with particular attention to what is used as the reference. 

4. It would be good if Figure 1 could be represented as a dot/scatter plot vs a bar graph tp see individual data points. The data points could even be connected and paired T tests performed rather than non-paired.

5. In Figure 2, its unclear what the statistical differences between the groups were?  The anavoa did indicate a difference, but then what did post-hoc analysis reveal about where the difference is?

6. In Figure 3, you show the decline in antibody levels between T3-1 and T3-2.  Why did you not do a paried T test here again?  What you show is not a rate, but rather a %loss.  A suitable statistical approach should be applied here 

7.  In Figure 4, it is uncluer what the indicated p-values are for? What was being tested here? Was it another Anova?? Also,  the authors indicate in the study limitations within the discussion that there was a loss of follow up of the participants from T3-1 to T3-2 and T3-3. What are the numbers? The numbers indicated in Figure 4, then only apply for T3-1? 

8.  Unclear of the statement being made on line 154-155. Is there an error?

Also, the language on line 156 is unclear, "However, the increase in antibody level was reduced". What is this statement refering to?

In line 164, it says that participants in the study received two doses of different vaccines as the primary course, but not all did as some only got moderna or BTN as the first 2 doses.

9. Another limitation to mention in addition to not measuring T cells, the study did not measure neutralization. 

10. In the final statement of the discussion, on line 199, perhaps the results suggest a 4th dose is needed to sustain higher circulating antibody levels against RBD, but its not clear if those will counteract emerging variants as stated. 

Author Response

Response to Reviewers

Response to Reviewer: 1

  1. With regards to the study participants, there are no inclusion or exclusion criteria included in the materials. Perhaps they are listed in previous publications, but at the minimum, it needs to be clear if the study excluded participants with immune suppressing conditions (HIV, Diabetes, kidney disease, pregnancy) or medications. 

Response)

Authors added inclusion and exclusion criteria in in section 2.1. (Line 67-69) as follows: The inclusion criteria were as follows: HCW with vaccination plan and consented HCW. Subjects who had COVID-19-related symptoms at the time of the study or pregnancy were excluded.

  1. In section 2.2, it states that the study was designed to detect RBD and neutralizing antibodies in serum, yet the study did not in fact did not assess any functional capacity of the antibodies, such as neutralization. This needs to be clarified.  Also, in this section, it should be made clear that ancestral RBD is utilized for the assay.

Response)

Authors clarified these points in section 2.2. (Line 87-91) as follows: The ancestral RBD was utilized for the assay. Plaque reduction neutralization (PRNT) are used to quantify the titer of neutralizing antibodies for a virus. A positive percent agreement study was performed with the SARS-CoV-2 IgG II Quant assay that were demonstrated to be positive (≥ 1:20) using a PRNT by the Broad Institute.

  1. In section 2.3. there is no description of the regression models used in the study, with particular attention to what is used as the reference. 

Response)

Authors described the regression models used in this study in section 2.3. (Line 106-109) as follows: The log-transformed values were used in regression models for anti-SARS-CoV-2 S-RBD IgG levels with a skewed distribution. Univariable and multivariable liner regression analyses were performed to verify the associated factors for anti-SARS-CoV-2 S-RBD IgG levels at 1 month after the third dose.

  1. It would be good if Figure 1 could be represented as a dot/scatter plot vs a bar graph tp see individual data points. The data points could even be connected and paired T tests performed rather than non-paired.

Response)

Authors depicted revised Figure 1 and conducted the paired t-test. Furthermore, authors described in section 2.3. (Line 102-104) as follows: The differences in antibody levels before and after the third vaccine dose were analyzed using paired t-test.

  1. In Figure 2, its unclear what the statistical differences between the groups were?  The anavoa did indicate a difference, but then what did post-hoc analysis reveal about where the difference is?

Response)

Authors depicted revised Figure 2 and marked results of post-hoc analysis.

  1. In Figure 3, you show the decline in antibody levels between T3-1 and T3-3.  Why did you not do a paried T test here again?  What you show is not a rate, but rather a %loss.  A suitable statistical approach should be applied here 

Response)

Authors tried to show the percent reduction at T3-3 compared to T3-1 and did a paired t-test. We depicted the revised Figure 3 and described in 2.3. section (Line 104-105) revised legend.

  1. In Figure 4, it is uncluer what the indicated p-values are for? What was being tested here? Was it another Anova?? Also, the authors indicate in the study limitations within the discussion that there was a loss of follow up of the participants from T3-1 to T3-2 and T3-3. What are the numbers? The numbers indicated in Figure 4, then only apply for T3-1? 

Response)

Authors analyzed mean difference of anti-SARS-CoV- S-RBD IgG antibody levels among the four vaccination schemes at T1-1, T2-3, T3-1, and T3-3 using ANOVA test. We described in Figure 4 legend (Line 158-164) as follows;

Figure 4. Mean anti-SARS-CoV-2 S-RBD IgG antibody levels for the four vaccination schemes before and after the vaccinations.

T0 (day of first dose); T1-1 (1 month after first dose); T1-3 (3 months after first dose); T2-1 (1 month after second dose); T2-3 (3 months after second dose); T3-1 (1 month after third dose); T3-3 (3 months after third dose).

The mean difference among four vaccination schemes was tested using ANOVA at T1-1, T2-3, T3-1 and T3-3, respectively. ** P values (p<0.001) were calculated using ANOVA test.

  1. Unclear of the statement being made on line 154-155. Is there an error?

Response)

Authors revised this point in Discussion section (Line 190-193) as follows: The antibody titer after homologous Moderna vaccine boosting was substantially higher in those who received the two primary Moderna vaccines than in those who only received two primary Moderna doses with heterologous booting [25]

-Also, the language on line 156 is unclear, "However, the increase in antibody level was reduced". What is this statement refering to?

Response)

Authors revised this statement in (Line 198-199) as follows: However, these increased antibody levels decreased as time goes on.

-In line 164, it says that participants in the study received two doses of different vaccines as the primary course, but not all did as some only got moderna or BTN as the first 2 doses.

Response)

Authors revised this statement in (Line 206-207) as follows: The participants in the present study received different vaccination schemes in the primary course.

  1. Another limitation to mention in addition to not measuring T cells, the study did not measure neutralization. 

Response)

Authors added the limitation that neutralization did not be measured in this study in (Line 231-232) as follows: and anti-SARS-CoV-2 S-RBD IgG antibodies were used as neutralizing antibody surrogates in this study.

  1. In the final statement of the discussion, on line 199, perhaps the results suggest a 4th dose is needed to sustain higher circulating antibody levels against RBD, but its not clear if those will counteract emerging variants as stated. 

Response)

Authors added the statement in conclusion (Line 251-253) as follows: These results suggest that another improved booster dose (a fourth dose) which would be able to counteract SARS-CoV-2 variants is needed.

Reviewer 2 Report

The study by Nah et al. is a prospective observational study that estimated the antibody response and persistence of immunogenicity after a third dose of BNT in homologous (ChAd/ChAd, BNT/BNT, and Moderna/Moderna) and heterologous (ChAd/BNT) vaccinations of two primary doses in different vaccination schemes. The third BNT vaccine dose vaccine was administered at a mean of 4 months after the second dose, and induced increased humoral responses in the various vaccination schemes, which were more prominent when the primary two doses were homologous mRNA vaccines. In practical terms, this study is an attempt to evaluate the antibody response after the third dose of the Pfizer vaccine against COVID-19. The study uses data from a cohort that has already been designed and used for another purpose. Thus, there are a number of weaknesses that need to be better explained and analyzed before the study is accepted.

Major revisons:

1) Need to improve the conclusion of the study in the abstract;

2) There is a large difference in the number of participants in each group that can compromise the results. The groups are also not homogeneous in terms of gender and age. How did the authors go about normalizing all this data?

3) What is the T2-3 group shown in Figure 1? What are the T1-1, T1-3, T2-1, and T2-3 groups shown in Figure 4? Authors need to explain the denomination of these groups in the Material and Methods and in the legends of the figures.

4) The results in table 2 are compromised with the difference in the number of participants in each group as questioned in question 2 above.

5) The most critical point of the study is that only the antibody response was evaluated and not the T-cell response, which also compromises the study's conclusions.

Author Response

Response to Reviewer: 2

1) Need to improve the conclusion of the study in the abstract;

Response)

Authors revised the conclusion of the study in the Abstract section as follows: The anti-SARS-CoV-2 S-RBD IgG levels at T3-1 were significantly associated with having received mRNA vaccines as the two primary doses (p<0.001). The third dose of BNT induced an increased humoral immune response in various vaccination schemes, which was more prominent for the two primary doses of homologous mRNA vaccines. However, this immunogenicity decreased within 3–10 months after the third dose. These results suggest that another booster dose (a fourth dose) which would be able to counteract SARS-CoV-2 variants is needed.

2) There is a large difference in the number of participants in each group that can compromise the results. The groups are also not homogeneous in terms of gender and age. How did the authors go about normalizing all this data?

Response)

Authors explained this limitation in Discussion section (Line 241-248) as follows:

This study was not randomized control study but prospective observational study in consented HCW cohort. There were vaccine supply shortages and interruptions in beginning of vaccination in Korea. The Korean government recommended various vaccine types and cross-platform mixed-dosing strategies according to vaccine supply. Koreans could not choose vaccine types. Nevertheless, authors tried to adjust for age, sex and vaccine types on analyzing data to overcome this limitation

3) What is the T2-3 group shown in Figure 1? What are the T1-1, T1-3, T2-1, and T2-3 groups shown in Figure 4? Authors need to explain the denomination of these groups in the Material and Methods and in the legends of the figures.

Response)

Authors explained the denomination of groups in Material and Methods (Line 72-78) and the legends of the Figures as follows: HCWs had been tested for anti-S-RBD IgG antibodies at T0 (day of first dose), T1-1 (1 month after first dose), T2-0 (day of second dose), T2-1 (1 month after second dose), and T2-3 (3 months after second dose). The findings of blood tests performed at three time points were analyzed as the antibody response after the third vaccination: T3-1 (1 month after third dose), T3-3 (3 months after third dose), and T3-4~10 (4~10 months after third dose).

4) The results in table 2 are compromised with the difference in the number of participants in each group as questioned in question 2 above.

Response)

Authors tried to adjust for age, sex and vaccine types on analyzing data using multivariable linear regression analyses to overcome this limitation.

5) The most critical point of the study is that only the antibody response was evaluated and not the T-cell response, which also compromises the study's conclusions.

Response)

Authors described this limitation in Discussion section (Line 234-236) as follows: we only evaluated the anti-SARS-CoV-2 S-RBD IgG levels after the third vaccination dose, which would compromise the study conclusions.

Thank you very much!

Round 2

Reviewer 2 Report

The authors answered all reviewers' questions and made the necessary changes in the new version of the manuscript, which can now be accepted for publication in Viruses journal.